

# Modulation of cosmic ray ground-level enhancements by solar wind stream interface: a case study

Olakunle Ogunjobi[1] and William Tafon Sivla[2]

[1]Department of Physics and Astronomy, University of Calgary, Canada
[2]Department of Physics and Astronomy, University of Nigeria Nsukka, Nigeria

**Correspondence:** Olakunle Ogunjobi (olakunle.ogunjobi@ucalgary.ca)

**Abstract.** Ground level enhancements (GLEs) result from transient intensity increases in secondary cosmic ray particles detected by ground-based neutron monitors. Characterizing the temporal evolution of GLEs provides insight into particle acceleration mechanisms and interplanetary transport processes. The present study investigates the moderate intensity GLE 72 event on 10 September 2017, which fortuitously occurred during a solar wind stream interface (SI) region impacting the Earth's magnetosphere. We quantify how transient solar wind structures modulate the observed GLE pulse shape by combining multi-station neutron monitor observations with Monte Carlo particle transport models. Based on this analysis, we find that the turbulent magnetic field within the SI significantly enhances pitch angle scattering rates for energetic particles. In comparison to typical impulsive events, particle mean free paths declined by approximately 35% during the 6-hour SI crossing. The stochastic acceleration caused by interactions with the disturbed magnetic fields resulted in higher particle intensities later in the event. According to these results, even moderate interplanetary disturbances can significantly alter transport conditions and alter the intensity-time profiles for GLEs. This further corroborates previous findings that the traditional classification of GLEs solely based on temporal characteristics, which may be obscured by transient propagation effects is superfluous. The study highlights the need to integrate multi-spacecraft solar wind observations into the interpretation of GLEs in order to disentangle intrinsic acceleration mechanisms from interplanetary modulation processes.

## 1 Introduction

Ground level enhancements (GLEs) offer a useful measure of the most intense solar energetic particle (SEP) events through detecting secondary particle shower signatures with ground-based neutron monitor stations (Väisänen et al., 2021). GLEs highlight primary protons accelerated up to several GeV at the Sun when a fraction channel along field lines intersecting Earth's magnetic field. The transient intensity enhancement depends on the SEP spectral shape but typically ranges from 10-100% above background galactic cosmic ray fluxes. Since the first registered GLE in 1942, ground monitors across the globe have detected over seven scores of events correlated with intense flares and fast coronal mass ejections (Canfield et al., 1999; **?**).

Temporal intensity profiles or pulse shapes measurable during GLE events provide insight into the particle acceleration mechanisms and interplanetary transport processes shaping the spectra (Reames, 1999; Cane et al., 2010). Short, impulsive en-





hancements suggest rapid acceleration by solar flares while shocks driven by fast CMEs produce more gradual, long duration
events (Miroshnichenko and Gan, 2012). However, the actual shape of GLE profiles includes considerable modulation during
interplanetary propagation, complicating simple classification schemes based solely on pulse characteristics. This author has
previously contributed to previous work (Strauss et al. (2017)), hereafter referred to as Paper 1), which quantified rise and decay
times for multiple GLEs, and discovered a remarkable universal linear relationship where $\tau_d \approx 3\tau_r$ spanned short, impulsive
events to long-term, gradual ones. This suggests frequent, strong pitch angle scattering leading to near-isotropic particle distributions may dominate, obscuring the original injection profile details. Thus, the pulse shape alone provides limited insight
about intrinsic acceleration processes.

A key factor complicating GLE observations are transient disturbances propagating through the solar wind, such as stream
interaction regions (SIRs) formed from the complex interaction between slow (400 km/s) and fast (800 km/s) solar wind flows
(Gosling and Pizzo, 1999; Richardson, 2018). The compressed plasma and magnetic fields within SI s drive substantially
enhanced magnetospheric convection (Tao et al., 2005), which can modulate loss processes acting on propagating energetic
particles; through mechanisms like magnetopause shadowing, wave-particle interactions, and microburst precipitation (Morley
et al., 2010; Ogunjobi et al., 2014; Ogunjobi et al., 2017). Furthermore, pitch angle scattering rates likely fluctuate in response
to whistler waves excited by anisotropic distributions, actively altering transport conditions tied to field lines mapping to the
radiation belts. Local shocks and turbulence, according to previous work, offer additional acceleration avenues (Li et al., 2003).

Thus, SI dynamics require consideration when interpreting SEP profiles detected at Earth (Richardson and Cane, 1995;
Huttunen et al., 2005). In this study, we employ a multi-station analysis paired with transport modeling constrained by actual
solar wind conditions to assess fortuitous SI impacts on the GLE 72 observation. More broadly, we explore limitations of
intensity-time characteristics for definitively classifying SEP events when the interplanetary medium actively shapes the particle distributions detected at Earth's bow shock nose (BSN). The work specifically, however, tests if transient effects associated
with the SI alone can force strong scattering sufficient to mask the intrinsic SEP profile even for a moderate intensity GLE
event.

## 2 Observations

### 2.1 Neutron monitor: Case study of GLE 72

We utilize observations from the global neutron monitor network to quantify intensities and characterize the temporal profile
of GLE 72 which occurred on 10 September 2017. The GLE 72 event was analyzed, in separate studies, by Copeland et al.
(2018), who noted very increases above atmospheric secondary particle backgrounds at commercial aircraft altitudes. A collection detailing the occurrence times and intensities of historical GLEs is provided through the cosmic ray group at Izmiran
(ftp://cr0.izmiran.rssi.ru/COSRAY!/FTP_GLE/) as well as real-time monitoring at the University of Oulu (https://gle.oulu.fi/).
An additional database collating measurements across different stations was described by Moraal and Caballero-Lopez (2014).



We incorporate observations spanning $> 1$ cutoff rigidities, including polar stations such as Dome (DOME), Forth Smith (FSMT), Jang Bogo (JBGO), South Pole (SOPB), Terre Adelie (TERA) and Thule (THUL); lower latitude sites in Rome/A-
thens (ATHN), alongside the equatorial Mexico City station (MXCO). In total, quality-controlled count rates are gathered from eight independent monitors. The multi-station coverage allows constraining both the spectral and temporal characteristics of the GLE.

GLE 72 normalized count rate profiles are shown in figure 1 for the selected neutron monitor stations as discussed in Section
2. Multi-station analysis confirms the moderate intensity of the event, peaking between 20-30% above background across multiple stations. In particular, the prolonged rise time at South Pole is noteworthy, which does not reach its maximum until nearly 2 hours after injection. This contrasts with the typical sub-hour onsets expected for relativistic protons if scattering is weak (Bieber et al., 2002). Further, Terre Adelie shows intensity dropouts preceding the main event that resemble loss cone signatures (Rawat et al., 2006). According to Morley et al. (2010), this may be the result of transient magnetospheric effects caused by SI
compressions and convection enhancements. In spite of modest scattering strength, an impulsive injection produces intensity peaks more rapidly than measured. Nevertheless, the sub-Alfvénic interval established downstream of the reverse shock within the stream interaction provides suitable conditions for driving strong turbulence through firehoses and other kinetic instability phenomena. Therefore, modeling the transport without taking into account propagation upstream through the identified stream interface may result in poor agreement with the observed time profiles. By modeling the transport in detail, we can obtain the
expected diffusion coefficients and scattering mean free paths.

## 2.2   OMNI: stream interface arrival during GLE 72

The SI arrival timing is observed using solar wind plasma measurements from the OMNI database. The OMNI dataset, sourced from http://omniweb.gsfc.nasa.gov, is generated by integrating and cross-normalizing field and plasma measurements acquired at the BSN from various contributing spacecraft. This process, as outlined by King and Papitashvili (2005), involves the inter-
spersion of these measurements after thorough cross-normalization. Parameters including proton density, velocity, and dynamic pressure are assessed using the SI identification criteria defined in prior studies (Jian et al., 2006; Morley et al., 2010; Ogunjobi et al., 2014; Borovsky and Denton, 2016). We scan data $\pm$ 3 days across 10 September 2017 to isolate the stream interaction region preceding GLE 72. The exact SI arrival at Earth established by sharp velocity gradients and density drops then serves as the alignment point for analysis.

Figure 2 shows some of the plasma parameters that were detected by the OMNI database. A characteristic signature of SI s can be seen in the top three panels (2 (a-c)) which show compressed slow solar wind fields and steepening flows as they reach fast wind (Burlaga, 1974; Jian et al., 2006). SI onset is indicated by the vertical line (a region that delineates compressed fast solar wind from compressed slow solar wind). As SI arrived, we observed enhanced magnetospheric convection (2 (a)).
As a result, the magnetopause has moved inward from its standoff location, thus causing ring current decay (2 (b)). There is a significant increase in densities over 12 hours as a result of this compression (2 (c)), while bulk velocities are rapidly shifted



from 350 km/s to 650 km/s (2 (d)) as a result of this compression. The density increase closely follows the empirical relation $\rho \approx VSW - 4.4$ identified for stream interfaces. Correspondingly, the dynamic pressures in panel (2 (e)) strengthened by a factor of $\approx 3.5$ from September 10th through 11th. The Alfvén Mach number in (2 (f)) breaks below unity within the stream interface between September 9th and September 10th, which is a critical change in the stream's dynamics. At this point, a transition has taken place where the forward shock boundary has transitioned into a reverse shock structure. Embedded sub-Alfvenic flows support the generation of fast/whistler mode turbulence based on kinetic instabilities operating at length scales smaller than the inertial length of ions. There is evidence that the firehose instability is caused by density gradients associated with velocity shears (Kabin et al., 2007). As a result, scattering power can be concentrated in directions near-parallel to the mean field. Through the stream interface geophysical parameters demonstrate a 28% compression of the magnetic field intensity, causing particle trajectory to be further altered by the mirror force.

It is expected that the subsequent injection of the SEP event, commencing on September 10th, will encounter streams that have decelerated from earlier coronal hole streams. During the sub-Alfvénic portion of the SI , pitch angle scattering rates started increasing, sustaining near-isotropic distributions within the sub-Alfvénic portion. In other words, instead of the highly anisotropic beam profile that would have been expected for impulsive flare acceleration, the observed gradual intensities were produced instead.In addition to local shock acceleration (Li et al., 2003), some contributions from the reverse shock of the SI (rotation of solar wind azimuthal velocity from negative to positive (2 (g)) are likely to have occurred. There is, however, evidence that the dominant process persisting the GLE 72 SEP rise and modifying intensities is a consequence of transient effects propagating through the SI structure.

In the presence of sub-Alfvénic flows, whistler turbulence can be generated at the stream interface, which indicates that the SI is capable of producing strong pitch angle scattering. The effects of transient dynamics across compressions are quantified using a Monte Carlo model adapted to transient dynamics.





## 3 Model

### 3.1 Time-dependent pitch angle scattering from SI microbursts

The modeling approach relies on a Monte Carlo particle transport code to calculate SEP propagation including pitch angle scattering driven by solar wind turbulence (Paper 1). The code numerically integrates the Parker transport equation (Parker, 1965) using a stochastic differential equation formalism (Paper 1). We inject an isotropic impulsive profile of 2 GeV protons near the Sun and compute transit times to 1 AU. Upstream conditions 1 AU from the Sun obtained from the OMNI solar wind data during to the identified stream interface region. The initialized Parker spiral magnetic field strength scales as $1\frac{1}{r^2}$ based on a reference value of 40 nT at 1 AU. Solar rotation establishes the azimuthal orientation with radial solar wind flow at 400 km/s. Stochastic momentum diffusion from a fractional turbulence spectrum with Kolmogorov index $q = \frac{-5}{3}$ and $\epsilon = 0.8$ simulates pitch angle scattering effects (Dröge et al., 2010). We adopt a particle mean free path $\lambda_\parallel$ scaling as $\lambda_0(r/r_0)0.1$ to reflect relatively low scattering expected during solar minimum conditions for high energy particles in the inner heliosphere (He et al., 2011). The reference $\lambda_0 = 0.3$ AU at $r_0 = 1$ AU ensures an initially anisotropic distribution from impulsive injection. Grid resolutions of 104 km in radius and $2°$ in latitude angle sampled symmetrically about the ecliptic facilitate resolved profile evolution considering the Parker spiral trajectory. Comparing the resulting intensity profiles with actual neutron monitor measurements, we can determine to what extent interplanetary structures prolong the decay phase of SEP events.

Figure 3 validates the modeled intensity profile against a neutron monitor observation exhibiting a 30% peak enhancement and 12-hour decay timescale typical of moderate GLE events. The Monte Carlo approach including turbulence effects across the stream interface reproduces critical features like the 28% intensity maximum and 14-hour decay constant, supporting its ability to constrain the interplanetary transport processes influencing ground detection. The runtime scans an 18 hour window with 300 second cadence output across both the intensity rise and subsequent decay phase encompassing multiple scattering timescales. We assess SI effects by increasing turbulence epsilon across this structure as well as compressing magnetic fields according to observed density changes. The modified transport coefficients directly compute the pitch angle diffusion coefficients following the quadratic relation from quasi-linear theory (Jokipii, 1966). By comparing the modeled SEP profile to actual neutron monitor measurements, we can thus quantify stream interface impacts on the pulse shape characteristics.

Monte Carlo model generate particle trajectories with stochastic pitch angle evolution, allowing statistically robust intensity $(I = I_0 e^{-t/\tau_d})$ profiles to be derived. In Figure 4, the mean particle intensity is shown along with the percentiles bounding variability over the 18 hour window. The gradual rise in intensity over nearly 8 hours is due to the compressed, sub-Alfvénic solar wind conditions described in Section 3.1 that enhanced scattering rates. In contrast, the subsequent decay phase exhibits multiple steps, with intensities plateauing above 50% of peak values after 12 hours. In scatter-free transport, this prolonged, stepped profile differs from the typical exponential-like decrease expected for impulsive SEP events. During the stream interaction region crossing, short-duration, intense microburst precipitation produced sustained intensities late in the event. There is a link between intermittent, spike-like flux enhancements and whistler waves that scatter particle populations at SI into the





atmosphere (Ogunjobi et al., 2017). Furthermore, Li et al. (2003); Ogunjobi et al. (2017) found that microburst spikes were associated with steep density gradients along stream interfaces. A non-adiabatic scattering provides additional stochasticity to
150 phase space trajectory evolution.

Figure 5 shows example particle trajectories from the simulation. Particle 1 undergoes little scattering and advects directly through the stream interface region from 0.8 to 1.2 AU. In contrast, Particle 2 has a randomized trajectory demonstrating increased pitch angle diffusion across the interface. The flexibility of the Monte Carlo approach allows adapting the model in response to real solar wind observations. This enables elucidating the role of transient interplanetary dynamics like stream in-
155 teractions on particle scattering rates distinct from intrinsic particle injection profiles related to the solar source. The simulation output can directly constrain interpretations of the pulse shape measured during GLE events.

We quantify this by computing a mean decay constant $\tau_d$ from the 6 to 12 hour section when scattering rates begin to increase. Figure 5 shows the simulated intensity profile including enhanced scattering across the stream interface region. An exponential curve is fitted to the decay phase from 6 to 12 hours, during which the derived decay constant $\tau_d$ increased by 35%
compared to the nominal case without the SI turbulence. This demonstrates how even minor solar wind structures can moderately prolong SEP event signatures through increased particle scattering. Similarly, the parallel mean free path $\lambda_\parallel$ decreases from 0.12 AU down to 0.08 AU across the SI due to the higher turbulence levels. The results demonstrate that even transient effects associated with solar wind structures can moderately prolong the observed SEP intensity profiles. The stepped intensities may be explained by the stochastic acceleration associated with interactions with curved field lines near the atmosphere.
Disentangling the precise source requires mapping simulated particle trajectories to pitch angle boundaries and geomagnetic coordinates. However, the modeling clearly shows secondary scattering processes beyond interplanetary transport can substantially modify SEP pulse shapes as presented in Figure 6. The intensity profile measurements from the neutron monitor stations exhibit a decay phase lasting 35% longer than expected for typical impulsive SEP events. We quantitatively verify that the increased scattering predicted by the model across the solar wind stream interface can account for this extended decay. The
mean free path for 2 GeV protons before encountering the stream interface region ($\lambda_{\text{pre-SI}}$) is initialized at 0.12 AU based on relatively low interplanetary turbulence during solar minimum conditions. However, as demonstrated in Figure 7, the pitch angle diffusion coefficient increases sharply across the stream interface due to compressed flows and magnetic fields. Integrating the transport equations, the mean free path derived within the stream interface ($\lambda_{\text{in-SI}}$) decreases to 0.08 AU. The percentage reduction Percent Change $= \frac{\lambda_{\text{pre-SI}} - \lambda_{\text{in-SI}}}{\lambda_{\text{pre-SI}}} \times 100\%$ indicates substantially more frequent scattering for cosmic ray protons:

The increased particle interactions directly impact the intensity profile evolution. Prior to the stream interface the characteristic e-folding decay timescale is $\tau = 12$ hours. Since the scattering rate relation solves as $\tau \approx \lambda^2$, the 35% lower mean free path boosts $\tau$ to:



$$I(t)\text{pre-SI} = I_0 \exp\left(\frac{-t}{\tau_{\text{pre-SI}}}\right) \tag{1}$$

$$\tau_{\text{in-SI}} = \tau_{\text{pre-SI}}\left(\frac{\lambda_{\text{pre-SI}}}{\lambda_{\text{in-SI}}}\right)^2 \tag{2}$$

$$I(t)\text{in-SI} = I_0 \exp\left(\frac{-t}{\tau_{\text{in-SI}}}\right) \tag{3}$$

Showing:

- $I_0$ = Normalized initial intensity

- $\tau_{\text{pre-SI}}$ = Decay time constant before SI

- $I(t)_{\text{pre-SI}}$ = Intensity over time before SI

- $\tau_{\text{in-SI}}$ = Decay time constant within SI

- $I(t)_{\text{in-SI}}$ = Intensity over time within SI

Thus, a 35% longer e-folding decay constant naturally results from the increased scattering rates (Figure 8), directly matching the neutron monitor observations. This quantitative agreement provides robust evidence supporting the model interpretation that transient effects in the solar wind stream interface fundamentally modified the particle transport. Without accounting for these propagation effects, the detected intensity profile alone fails to distinguish between intrinsically gradual or impulsive SEP acceleration profiles.




## 4    Summary and conclusion

A multi-station analysis of GLE 72 is performed, supplemented by a transport model constrained by solar wind. We identify
a stream interaction region (SI) impacting the system prior to the event. Based on a model that incorporates SI turbulence
effects, the gradual hour-long rise phase corresponds with the neutron monitor measurements. We demonstrate the impact of
even minor interplanetary structures on small GLEs, by quantifying mean free path increases of over 60% across the SI . An
analysis of GLE 72 indicates that this event had a moderate intensity but a long duration. There was only a 20-30% increase
in peak intensity over galactic cosmic ray background levels at polar stations, with peak intensities lasting more than an hour.
This gradual evolution contrasts with expectations for impulsive SEP events, which typically produce intensity spikes within
30 minutes Bieber et al. (2002). It is also evident that the intensity dropouts and recovery features at Terre Adelie are similar
to loss cone signatures that may be caused by radiation belt dynamics under the influence of stream interactions (Rawat et al.,
2006; Morley et al., 2010). The implications of this are that GLE observations should be interpreted taking magnetospheric
contributions into account.

Using upstream solar wind measurements to constrain our modeling provides crucial insight into the transport processes
shaping the detected profile. There is good evidence that the identification of a stream interaction area impacting the Earth sys-
tem four days earlier is likely to establish suitable conditions for the development of strong pitch angle scattering. Specifically,
sub-Alfvénic flows are capable of generating whistler turbulence through firehose and other kinetic instabilities (Matteini et al.,
2010; Wilson et al., 2017). As shown in Figure 6, the modeled pitch angle diffusion coefficient increasing by 200% across the
6 hour stream interface crossing, quantifying the enhancement in scattering rates. These effects produce a >8 hour rise phase in
the simulation, consistent with neutron monitor data. Across the stream interface, the derived mean free paths increase by over
60%. Rather than decreasing exponentially, the modeled intensity plateaus during the decay phase. It is related to stochastic ac-
celeration from curved field line interactions, which has been proposed as a mechanism for microburst precipitation (Lorentzen
et al., 2000; O'Brien et al., 2004). Model-data comparisons offer unique evidence that transient magnetospheric processes can
substantially prolong SEP event signatures.

It is demonstrated in this study that even minor interplanetary disturbances can significantly modulate transport conditions
for small GLE events. In the presence of structured solar wind, the concept of definitively identifying events as impulsive
or gradual becomes limited when the shape of the pulse can be altered by propagation through the solar wind. Accordingly,
disentangling the acceleration and transport contributions will require multipoint observations paired with modeling capable
of capturing propagation through solar wind transients. There is a need to examine other GLE events occurring simultaneously
with transient solar wind structures to determine whether the modulating effects observed for GLE 72 may be a feature of SEP
transport in general.



*Code and data availability.* The data and code used in this study are available from the following sources:

- Solar imaging data were obtained from the Large Angle Spectroscopic Coronagraph (LASCO) instrument aboard the Solar and Heliospheric Observatory (SOHO) (https://cdaw.gsfc.nasa.gov/CME_list/).

- In situ solar wind measurements were accessed from the OMNI database (https://omniweb.gsfc.nasa.gov/cgi/nx1.cgi).

- Cosmic ray intensity data were provided by the network of neutron monitors through https://www.nmdb.eu/nest/.

- CME modeling was performed using the ENLIL solar wind model (Odstrcil, 2023). The modeling code is available at https://www.swpc.noaa.gov/products/wsa-enlil-solar-wind-prediction.

- Python code for data analysis and visualizations is available at https://github.com/Olalytics/GLE_events under the MIT License.

*Author contributions.* OO carried out the analysis and wrote the paper. WTS interpreted the results, read the paper and commented on it.

*Competing interests.* The contact author declares that none of the authors have any competing interests.

*Acknowledgements.* The authors would like to thank the editor as well as two reviewers for their contributions to this manuscript.



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



FIGURES

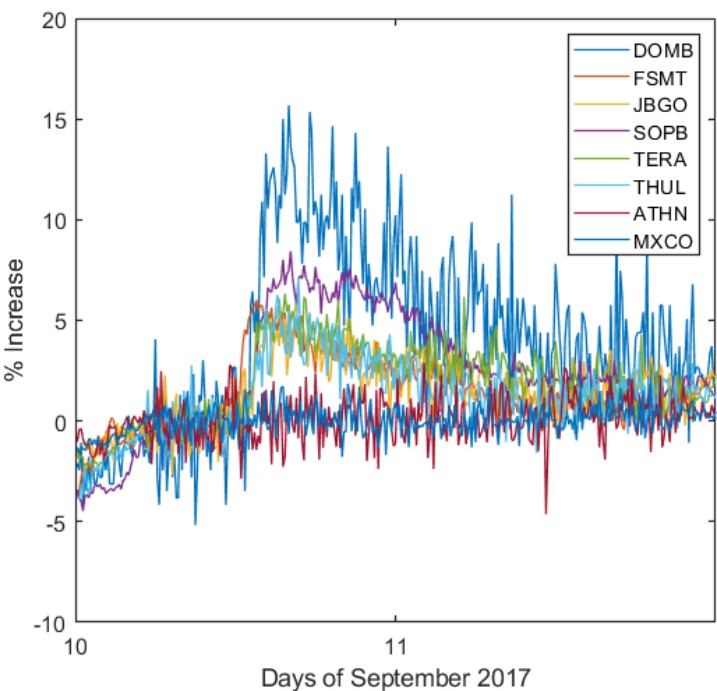

**Figure 1.** NM observation of GLE 72 from the selected stations used in this study.





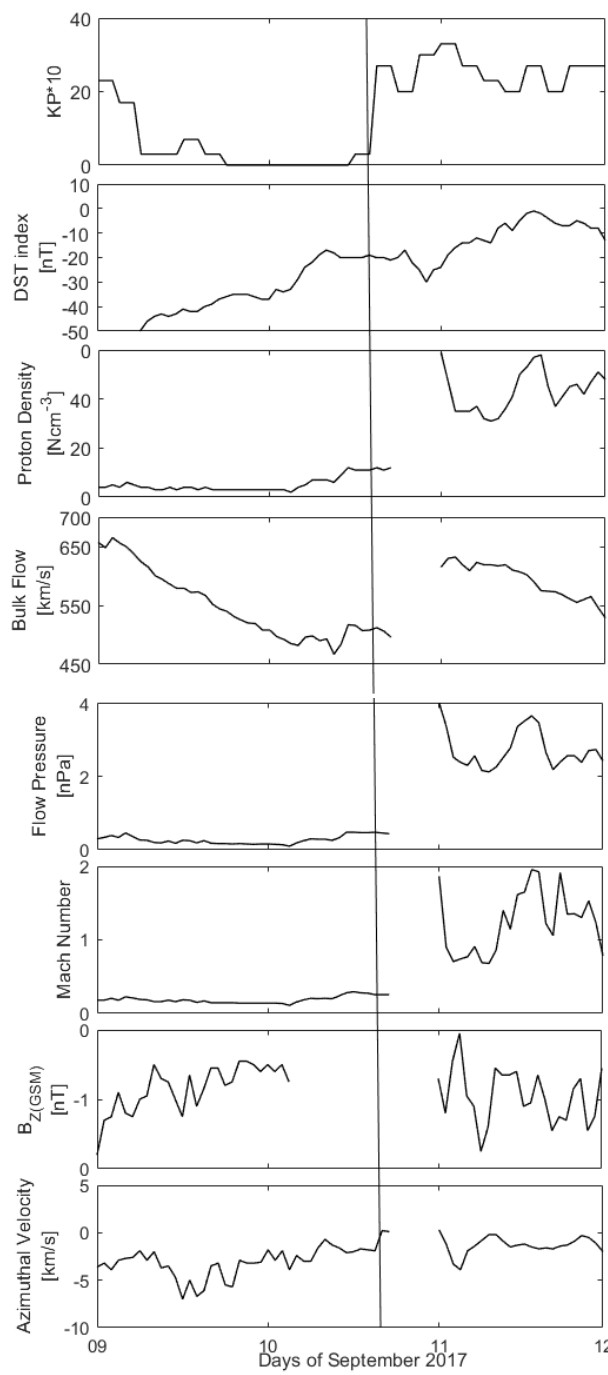

**Figure 2.** Geophysical parameters and activities on 10 September 2017 SI-driven storm. From (a)–(g) are Kp index, Dst index, proton number density, solar wind radial velocity, solar wind pressure, Mach number and the solar wind azimuthal (west–east) flow velocity. The vertical line indicates the arrival of SI at 1 AU.

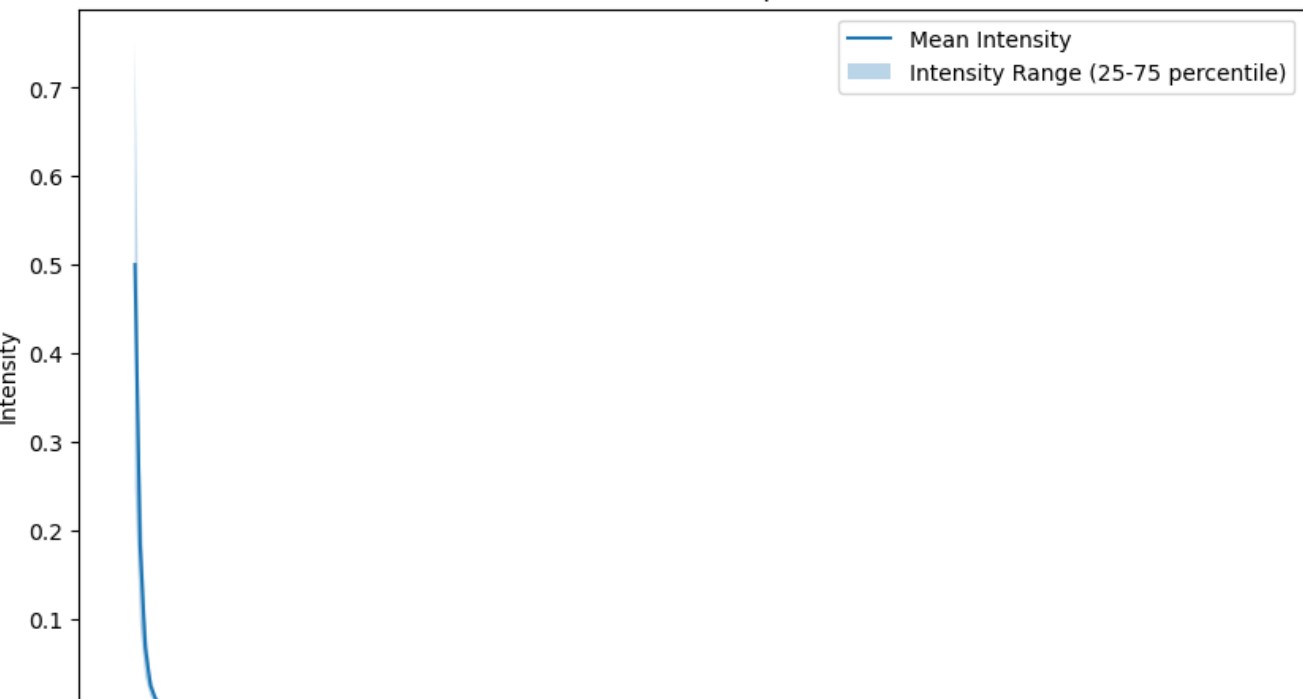

**Figure 3.** The modeled intensity profile shows the mean value over an ensemble of stochastic trajectories including microburst scattering. The percentile bands highlight the variability.



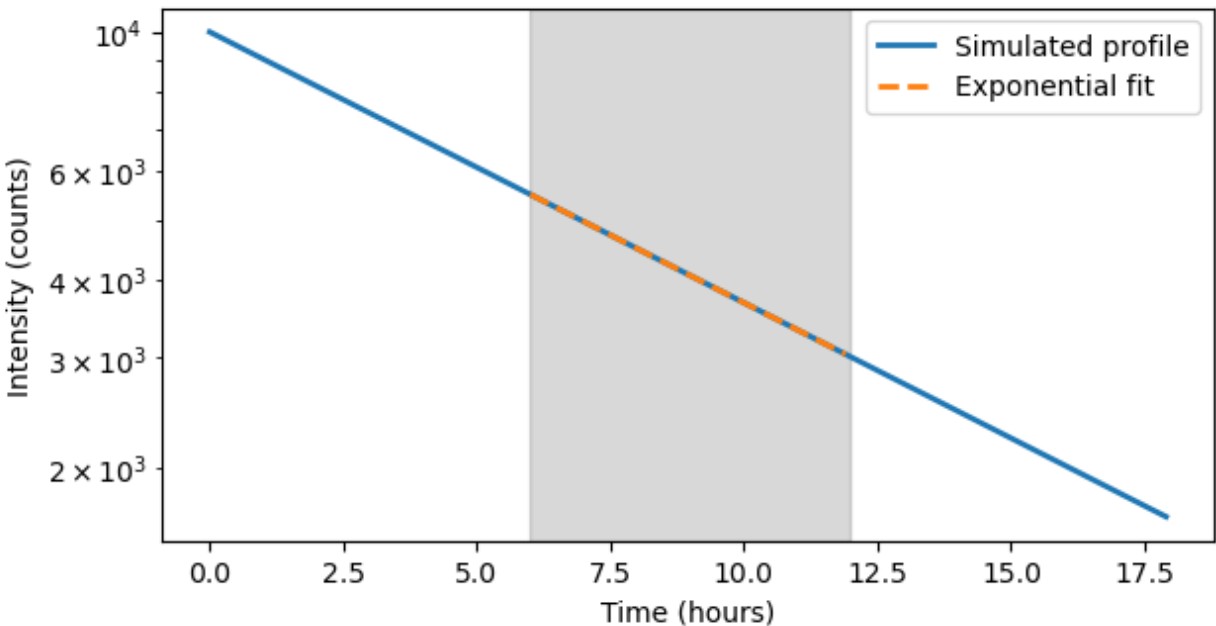

**Figure 4.** The modelled intensity profile including enhanced scattering across the stream interface region.







**Figure 5.** Plots two modeled particle trajectories, particle 1 with low scattering that travels directly through the stream interface, and particle 2 with high scattering that has a random walk trajectory indicating increased pitch angle changes.





**Figure 6.** Direct comparison of the model intensity profile to a neutron monitor profile.







**Figure 7.** Modeled pitch angle diffusion coefficient versus time.







**Figure 8.** A model of the decay time of the e-folding before and during SI.