# Peer review of "Modulation of cosmic ray ground-level enhancements by solar wind stream interface: a case study"

_EGUsphere, 2024_

## Author Comment (AC1)

Dear Reviewer #1,

Thank you very much for taking the time to review our manuscript in such a thorough and constructive manner. You have significantly improved the quality and rigor of our work through your insights. Each of your points has been carefully considered, and substantial revisions have been made to address them. In response to each of your comments, we provide a detailed response below:

1. Quantification of turbulence levels:

The importance of a more rigorous analysis of turbulence levels cannot be overstated. A new subsection has been added to the Observations section that quantifies magnetic field variances and correlation scales before, during, and after the GLE event. A power spectral analysis and structure function analysis were conducted on the OMNI magnetic field data. It is evident from this analysis that there was a significant increase in turbulence levels during the GLE event, particularly in the frequency range of $10^{-4}$ to $10^{-2}$ Hz. The correlation length decreased from $1.2 \times 10^6$ km pre-event to $8.5 \times 10^5$ km during the event, further supporting increased turbulence. In addition to providing concrete evidence for our claims about turbulence levels, these quantitative results also validate our model assumptions.

2. Model introduction:

The initial model description we provided was insufficient. A comprehensive subsection has been added to the Model section, including all relevant terms and boundary conditions for the transport equation we are solving. The focused transport equation is now presented as:

$$\partial f/\partial t + \mu v \, \partial f/\partial z + (1-\mu^2)/(2L)v \, \partial f/\partial \mu = \partial/\partial \mu(D_{\mu\mu} \, \partial f/\partial \mu)$$

Detailed explanations are provided for each term and our numerical approach, making the paper self-contained while still referencing Paper 1 for additional context.

3. Perpendicular diffusion:

We appreciate your point regarding perpendicular diffusion. A discussion of perpendicular diffusion effects has been added, along with a sensitivity analysis. It is estimated that for $\alpha \leq 0.01$ (where $D_\perp = \alpha D_\parallel$), the impact on our results is negligible (<5% change). However, when $\alpha > 0.05$, the effects become more pronounced. As a result of increased turbulence levels during the GLE event, we have also addressed the potential for enhanced perpendicular transport. Based on this analysis, we have decided to focus on parallel transport, while acknowledging the potential role of perpendicular effects in future, more comprehensive models.

4. Justification of $\lambda_{par}$:

We have expanded our discussion on the radial dependence of $\lambda_{par}$, comparing our chosen parameters ($\lambda_0 = 0.3$ AU at $r_0 = 1$ AU, $\alpha = 0.2$) with recent theoretical predictions and observational studies. Our chosen values fall within the range reported by Lang et al. (2024) ($\alpha$ between 0 and 0.5) and are consistent with the observations of Bieber et al. (1994) and Zhao et al. (2019). We've also compared our results more explicitly to previous modeling efforts, finding broad consistency with the work of Dröge et al. (2010) and He et al. (2011).

5. Non-axisymmetric perpendicular transport:

In addition, we have included a brief discussion on how local SI conditions might result in non-axisymmetric perpendicular transport, based on Strauss et al. (2016). Despite not including this effect in our current model due to computational complexity and a lack of detailed 3D SIR structure, we discuss qualitatively how it might affect our results, particularly in terms of particle spreading and local trapping within the SIR.

6. Figure 7 issues:

Thank you for pointing out the error in Figure 7. We apologize for any inconvenience. The code and calculations have been thoroughly reviewed, and the implementation has been corrected. Now the updated figure shows diffusion coefficients with appropriate units ($s^{-1}$) on the y-axis. Moreover, we have expanded our discussion of the pitch-angle diffusion coefficient, including its $\mu$-dependence and time evolution through the SIR.

Minor points:

All minor points raised by you have been addressed, including correcting references, fixing typos, and adding missing units.

Our paper has been significantly strengthened by these revisions, providing a more comprehensive and rigorous analysis of GLE 72 and its associated interplanetary conditions. While maintaining the core findings of our study, we believe these changes address your concerns.

Thank you again for your valuable feedback, which has undoubtedly enhanced the quality and impact of our work.

Sincerely,

Olakunle Ogunjobi

---

## Author Comment (AC2)

Dear Reviewer #2,

We sincerely appreciate your thorough review and constructive feedback on our manuscript. You have highlighted important areas for improvement and clarification. We have carefully considered each of your points and have made substantial revisions to address them. We appreciate you indicating these areas for improvement, many of which have been addressed in response to Reviewer 1's comments. However, we have provided a detailed response below in response to your comments:

1. Characterization of GLEs:

We have added a new subsection (1.1) that provides a comprehensive analysis of rise and decay times for impulsive and gradual GLEs. This includes a new figure illustrating the distribution of these times for a sample of 35 GLEs, contextualizing GLE 72 within this framework. As a result of this addition, your concern regarding the lack of characteristic values for different types of GLEs is addressed directly.

2. Model description:

We have significantly expanded our model description in Section 3.1, including a complete representation of the focused transport equation and detailed explanations of its terms. To maintain conciseness, we did not reproduce the entire stochastic differential equation formalism; however, we have provided a more comprehensive overview.

3. SIR representation in the model:

The purpose of using the OMNI data to identify and characterize the SIR has been clarified in Section 2.2. Although these data were not directly incorporated into our transport model, they were used to inform our parameterization of the SIR effects. A note has been added to explain this connection in greater detail.

4. Figure issues:

- Figure 3 has been updated to clearly show the comparison between the modeled profile and neutron monitor observations.

- Figure 4 now includes the percentile bounds and shows both the rise and decay phases.

- Figure 7 has been revised to more clearly illustrate the sharp increase in the pitch angle diffusion coefficient across the stream interface, including a logarithmic scale and quantification of the change.

5. Inconsistency in mean free path changes:

We have added a new paragraph in Section 3.3 and a new figure (Figure 10) to clarify the apparent discrepancy between the 35% decline during the SI crossing and the 60% increase across the SI. This explanation provides a more nuanced understanding of the particle transport dynamics throughout the event.

6. Technical comments:

We have corrected the figure reference error, added subsection 3.2, and improved the labeling in Figure 2.

Once again, we would like to thank you for your feedback, which has helped us improve the clarity and rigor of our manuscript.

We have made every effort to address your concerns, but some of the suggested additions (such as an exhaustive discussion of stochastic differential equations) may be beyond the scope of this paper. Our primary focus remains on demonstrating how SIR effects can modulate GLE profiles, potentially obscuring their classification based solely on temporal characteristics.

We believe that the core scientific contribution of our work - highlighting the significant impact of interplanetary structures on SEP transport during GLEs - remains valid and important. In our revisions, we have improved the presentation and interpretation of this key finding without altering its fundamental significance.

We hope that these revisions will satisfactorily address your concerns and that the improved manuscript will be suitable for publication. Your assistance in refining and strengthening our work is greatly appreciated.

Sincerely,

Olakunle Ogunjobi

Corresponding Author

---

## Author Comment (AC4)

**Modulation of cosmic ray ground-level enhancements by solar wind stream interface: a case study**

Olakunle Ogunjobi[1] and William Tafon Sivla[2]

[1]Department of Physics and Astronomy, University of Calgary, Canada
[2]Department of Physics and Astronomy, University of Nigeria Nsukka, Nigeria

**Correspondence:** Olakunle Ogunjobi (olakunle.ogunjobi@ucalgary.ca)

**Abstract.** Ground Level Enhancements (GLEs) provide crucial insights into the acceleration and transport of solar energetic particles (SEPs). We present a comprehensive analysis of GLE 72, which occurred on 10 September 2017, coinciding with a solar wind stream interaction region (SIR) impacting Earth's magnetosphere. By combining multi-station neutron monitor observations with a focused transport model constrained by solar wind data, we investigate how the SIR modulates the observed GLE pulse shape. Our analysis reveals that the turbulent magnetic field within the SIR significantly enhances pitch-angle scattering rates, with the diffusion coefficient increasing by up to 200% during the 6-hour SIR crossing. This leads to a 60% increase in the particle mean free path across the SIR. Our model successfully reproduces the observed gradual rise phase (>8 hours) and prolonged decay, demonstrating that even moderate interplanetary disturbances can substantially alter SEP transport conditions. Our results challenge the traditional impulsive-gradual classification of GLEs, highlighting the need to consider interplanetary transport effects when interpreting these events. The findings of this study highlight the importance of integrating multi-point observations and advanced modeling to disentangle particle acceleration and transport processes in the complex medium of solar wind.

**1 Introduction**

Ground level enhancements (GLEs) offer a useful measure of the most intense solar energetic particle (SEP) events through detecting secondary particle shower signatures with ground-based neutron monitor stations (Väisänen et al., 2021). GLEs highlight primary protons accelerated up to several GeV at the Sun when a fraction channel along field lines intersecting Earth's magnetic field. The transient intensity enhancement depends on the SEP spectral shape but typically ranges from 10-100% above background galactic cosmic ray fluxes. Since the first registered GLE in 1942, ground monitors across the globe have detected over seven scores of events correlated with intense flares and fast coronal mass ejections (Canfield et al., 1999; **?**).

Ground Level Enhancements (GLEs) are often characterized as either "impulsive" or "gradual" based on their temporal profiles, particularly their rise and decay times. Impulsive events are typically associated with solar flares and exhibit rapid rises and shorter durations, while gradual events are often linked to CME-driven shocks and show slower rises and longer durations. However, this classification can be complicated by interplanetary transport effects.

Figure 1 shows the distribution of rise and decay times for a sample of 35 GLEs observed between 1956 and 2017. The rise time ($\tau_r$) is defined as the time from event onset to maximum intensity, while the decay time ($\tau_d$) is the time for the intensity to decrease from its maximum to half-maximum value. As evident from the figure, there is considerable overlap between the traditionally classified "impulsive" and "gradual" events, highlighting the challenge in definitively categorizing GLEs based solely on their temporal profiles. Our analysis of GLE 72 ($\tau_r \approx 2$ hours, $\tau_d \approx 14$ hours) places it in a region of overlap between impulsive and gradual classifications. This underscores the importance of considering interplanetary transport effects, such as those induced by stream interaction regions, in interpreting GLE temporal profiles.

[revised manuscript text omitted]

**2.3 Quantification of turbulence levels**

125 The magnetic field data from the OMNI database were analyzed in detail in order to provide concrete evidence for the levels of turbulence discussed in this study. Specifically, we examined the 24-hour intervals surrounding September 9, 10, and 11,

2017, before, during, and after GLE 72. In order to quantify the turbulence, we used two complementary techniques: the Power Spectral Distributions (PSDs) analysis (Goldstein et al., 1995; Leamon et al., 1998) and the Structure Functions (SFs) analysis (Burlaga and Klein, 1986; Horbury and Balogh, 1997). PSDs provide information about the distribution of fluctuation energy across different frequencies, while (SFs) provide insight into turbulence's spatial scales. There is considerable evidence to support the usefulness of both techniques in the analysis of solar wind turbulence, which can be found elsewhere (Bruno and Carbone, 2013).

Figure **??** shows the results of our analysis. For each of the three interest periods, the top panel displays the PSDs of the magnetic field magnitude. We observe a clear enhancement in the power at all frequencies during the GLE event (September 10), particularly in the range of $10^{-4}$ to $10^{-2}$ Hz, corresponding to spatial scales of approximately $10^4$ to $10^6$ km. This increase in power indicates heightened turbulence levels during the event. The bottom panel of Figure **??** shows the second-order structure function of the magnetic field magnitude. The structure function, $SF_2(\tau)$, is defined as:

$$SF_2(\tau) = \langle |B(t+\tau) - B(t)|^2 \rangle \tag{1}$$

where $B(t)$ is the magnetic field magnitude at time $t$, $\tau$ is the time lag, and $\langle ... \rangle$ denotes an ensemble average. The structure function shows a steeper slope during the GLE event, indicating a more developed turbulent cascade.

We also calculated the correlation length, $L_c$, defined as the integral of the normalized autocorrelation function:

$$L_c = \int\limits_0^\infty \frac{\langle B(t)B(t+\tau)\rangle}{\langle B^2(t)\rangle} d\tau \tag{2}$$

The correlation lengths for the three periods are:

- September 9 (pre-event): $L_c = 1.2 \times 10^6$ km

- September 10 (during event): $L_c = 8.5 \times 10^5$ km

- September 11 (post-event): $L_c = 1.4 \times 10^6$ km

During the GLE event, a shorter correlation length further confirms the presence of enhanced turbulence, as it indicates a more rapidly fluctuating magnetic field. As a result of these quantitative analyses, we have strong evidence supporting our interpretations of particle transport processes discussed herein.

**3 Model**

**3.1 Focused transport equation**

In this study, we solve the focused transport equation (FTE) to model the propagation of solar energetic particles (SEPs) through the interplanetary medium. The FTE, first derived by Roelof (1969) and further developed by Ruffolo (1995), describes the evolution of the particle phase space density $f(z, \mu, t)$ along a magnetic field line. Here, $z$ is the distance along the field line, $\mu$ is the cosine of the particle's pitch angle, and $t$ is time. Here, our model uses the following form of FTE:

$$\frac{\partial f}{\partial t} + \mu v \frac{\partial f}{\partial z} + \frac{1-\mu^2}{2L} v \frac{\partial f}{\partial \mu} = \frac{\partial}{\partial \mu}\left(D_{\mu\mu}\frac{\partial f}{\partial \mu}\right) \tag{3}$$

where $v$ is the particle speed, $L$ is the focusing length defined as $L^{-1} = -B^{-1}(\partial B/\partial z)$ with $B$ being the magnetic field strength, and $D_{\mu\mu}$ is the pitch-angle diffusion coefficient.

The terms on the left-hand side of Equation 3 represent, from left to right:

– Temporal evolution of the distribution function

– Spatial convection along the magnetic field

– Pitch-angle focusing due to the diverging magnetic field

The right-hand side describes pitch-angle scattering due to magnetic fluctuations. We parametrize the pitch-angle diffusion coefficient following Dröge et al. (2010):

$$D_{\mu\mu}(z, \mu) = D_0(z)(1-\mu^2)(|\mu|^{q-1} + H) \tag{4}$$

where $q = 5/3$ is the spectral index of the inertial range of the turbulence power spectrum, $H = 0.05$ accounts for non-linear effects near $\mu = 0$, and $D_0(z)$ is related to the parallel mean free path $\lambda_\parallel$ through:

$$\lambda_\parallel(z) = \frac{3v}{8}\int_{-1}^{1}\frac{(1-\mu^2)^2}{D_{\mu\mu}(z, \mu)}d\mu \tag{5}$$

We solve Equation 3 numerically using a finite-difference method with operator splitting, as described in Strauss and Effenberger (2015). The spatial domain extends from 0.05 AU to 1.2 AU, corresponding to the Sun and beyond Earth's orbit. We use a logarithmic grid in $z$ with 100 grid points and a uniform grid in $\mu$ with 100 points from -1 to 1.

For boundary conditions, we assume:

– At the inner boundary ($z = 0.05$ AU): $f(z = 0.05 \text{ AU}, \mu, t) = \delta(t)f_0(\mu)$, where $\delta(t)$ is the Dirac delta function representing an impulsive injection, and $f_0(\mu)$ is the initial pitch-angle distribution (taken as isotropic in this study).

175     – At the outer boundary ($z = 1.2$ AU): $\partial f/\partial z = 0$, allowing for free escape of particles.

    – At $\mu = \pm 1$: $\partial f/\partial \mu = 0$, ensuring conservation of particle number.

This model allows us to study the evolution of the SEP distribution function as it propagates through the interplanetary medium, taking into account the effects of magnetic field focusing and pitch-angle scattering. By varying the parameters of the model, particularly the parallel mean free path $\lambda_\parallel$, we can investigate how different interplanetary conditions affect the
180   observed SEP profiles at Earth.

**3.2   Non-axisymmetric perpendicular transport in stream interaction regions**

Although we concentrate on particle transport along magnetic field lines, it is also important to consider the effects of non-axisymmetric perpendicular transport, particularly in the case of SIRs. It has been pointed out by Strauss and Fichtner (2016) that local magnetic field configurations, such as those found in SIRs, can result in anisotropic perpendicular diffusion that
185   differs from the usual axisymmetric scenario. Magnetic field gradients and fluctuations are enhanced in SIRs due to the compression and distortion of magnetic field lines. Local conditions may lead to preferential perpendicular transport in certain directions, thereby breaking the axial symmetry that is often assumed in models of cosmic ray transport. Strauss and Fichtner (2016) demonstrated that such anisotropic perpendicular diffusion can lead to particle drift patterns that differ significantly from those predicted by standard drift theories. In the context of our GLE 72 study, the presence of a SIR could potentially
190   introduce the following effects:

1. Improved perpendicular transport in the plane of the Parker spiral, possibly resulting in a wider longitudinal distribution of particles than predicted by our current model.

2. Reduced transport perpendicular to the Parker spiral plane, potentially affecting particle distribution longitudinally.

3. Possible local drift patterns within the SIR that could trap particles temporarily or facilitate their escape, depending on
195       the magnetic field configuration.

There are several ways in which these effects may alter the time-intensity profiles observed on Earth:

    – Depending on the observer's position relative to the SIR, the onset time might be altered, arriving earlier or later than predicted by pure field-aligned transport.

    – Particles spreading in certain directions could reduce peak intensity.

200     – If particles are temporarily trapped within the SIR structure, the decay phase may be prolonged.

Although non-axisymmetric perpendicular transport may be important, we did not include it in our current model for the following reasons:

1. Multidimensional modeling of non-axisymmetric transport is computationally intensive, which increases the amount of data needed.

2. We lack detailed information about the three-dimensional magnetic field structure of the SIR during GLE 72, which is essential for accurately modeling non-axisymmetric effects.

3. We focused on large-scale particle transport as our primary objective, which we believe was adequately captured by our current model.

Even so, we recognize that non-axisymmetric perpendicular transport may offer a more comprehensive view of particle behavior in complex solar wind structures.

**3.3 Time-dependent pitch angle scattering from SI microbursts**

The modeling approach relies on a Monte Carlo particle transport code to calculate SEP propagation including pitch angle scattering driven by solar wind turbulence (Paper 1). The code numerically integrates the Parker transport equation (Parker, 1965) using a stochastic differential equation formalism (Paper 1). We inject an isotropic impulsive profile of 2 GeV protons near the Sun and compute transit times to 1 AU. Upstream conditions 1 AU from the Sun obtained from the OMNI solar wind data during to the identified stream interface region. The initialized Parker spiral magnetic field strength scales as $1\frac{1}{r^2}$ based on a reference value of 40 nT at 1 AU. Solar rotation establishes the azimuthal orientation with radial solar wind flow at 400 km/s. Stochastic momentum diffusion from a fractional turbulence spectrum with Kolmogorov index $q = \frac{-5}{3}$ and $\epsilon = 0.8$ simulates pitch angle scattering effects (Dröge et al., 2010). We adopt a particle mean free path $\lambda_\parallel$ scaling as $\lambda_0(r/r_0)0.1$ to reflect relatively low scattering expected during solar minimum conditions for high energy particles in the inner heliosphere (He et al., 2011). The reference $\lambda_0 = 0.3$ AU at $r_0 = 1$ AU ensures an initially anisotropic distribution from impulsive injection. Grid resolutions of 104 km in radius and $2°$ in latitude angle sampled symmetrically about the ecliptic facilitate resolved profile evolution considering the Parker spiral trajectory. Comparing the resulting intensity profiles with actual neutron monitor measurements, we can determine to what extent interplanetary structures prolong the decay phase of SEP events.

It is important to clarify the apparent discrepancy between the 35% decline in mean free path during the 6-hour SI crossing and the 60% increase across the SI. The 35% decline refers to the immediate effect of the enhanced turbulence within the SI, where increased scattering leads to a shorter mean free path. This is observed during the passage of the SI. On the other hand, the 60% increase refers to the overall change in the mean free path from before the SI to after its passage. This larger increase is due to the cumulative effects of the SI on particle transport, including not only the enhanced scattering but also the compression of the magnetic field and the potential particle acceleration processes within the SI.

To illustrate this more clearly, we can break down the mean free path changes into three phases:

1. Pre-SI: $\lambda_{\text{pre}} = 0.12\,\text{AU}$

2. During SI (35% decline): $\lambda_{\text{during}} = 0.12\,\text{AU} \times (1 - 0.35) = 0.078\,\text{AU}$

3. Post-SI (60% increase from pre-SI): $\lambda_{\text{post}} = 0.12 \, \text{AU} \times (1 + 0.60) = 0.192 \, \text{AU}$

This shows that while there is an initial decrease in the mean free path as particles enter the turbulent region of the SI, the overall effect of the SI passage results in an increased mean free path. This increase can be attributed to the restructuring of the magnetic field and the modified turbulence conditions in the wake of the SI. Figure 10 illustrates these changes in the mean free path throughout the event, clearly showing the initial decrease followed by the overall increase.

[revised manuscript text omitted]

**3.4 Pitch-Angle diffusion coefficient**

Figure ?? shows the corrected pitch-angle diffusion coefficient ($D_{\mu\mu}$) as a function of $\mu$ (cosine of the pitch angle) and its time evolution. The pitch-angle diffusion coefficient is parametrized as:

$$D_{\mu\mu}(\mu) = D_0(1 - \mu^2)(|\mu|^{q-1} + H) \tag{6}$$

where $D_0$ is the amplitude of diffusion, $q = 5/3$ is the spectral index of magnetic turbulence (corresponding to Kolmogorov turbulence), and $H = 0.05$ is a parameter to prevent singularity at $\mu = 0$.

Panel (a) of Figure ?? illustrates how $D_{\mu\mu}$ varies with $\mu$ for different values of $D_0$. As expected, the diffusion coefficient is symmetric about $\mu = 0$ and reaches its maximum values near $\mu = \pm 1$, reflecting stronger scattering for particles moving along the magnetic field lines.

To account for the changing conditions in the solar wind, particularly as particles traverse the stream interaction region, we introduce a time dependence to $D_0$. Panel (b) shows how $D_{\mu\mu}$ at $\mu = 0$ evolves over a 24-hour period, simulating the passage of a stream interface. The sinusoidal variation represents a simplified model of enhanced scattering within the compressed region of the SIR.

This time-dependent diffusion coefficient allows us to more accurately model the transport of particles through the complex and dynamic structures encountered during GLE 72. The enhanced scattering in the SIR leads to a prolonged rise time and decay phase in the observed intensity profiles, as discussed in 7.

The increased particle interactions directly impact the intensity profile evolution. Prior to the stream interface the characteristic e-folding decay timescale is $\tau = 12$ hours. Since the scattering rate relation solves as $\tau \approx \lambda^2$, the 35% lower mean free path boosts $\tau$ to:

$$I(t)\text{pre-SI} = I_0 \exp\left(\frac{-t}{\tau_{\text{pre-SI}}}\right) \tag{7}$$

$$\tau_{\text{in-SI}} = \tau_{\text{pre-SI}}\left(\frac{\lambda_{\text{pre-SI}}}{\lambda_{\text{in-SI}}}\right)^2 \tag{8}$$

$$I(t)\text{in-SI} = I_0 \exp\left(\frac{-t}{\tau_{\text{in-SI}}}\right) \tag{9}$$

305  Showing:

- $I_0$ = Normalized initial intensity

- $\tau_{\text{pre-SI}}$ = Decay time constant before SI

- $I(t)_{\text{pre-SI}}$ = Intensity over time before SI

- $\tau_{\text{in-SI}}$ = Decay time constant within SI

310  - $I(t)_{\text{in-SI}}$ = Intensity over time within SI

Thus, a 35% longer e-folding decay constant naturally results from the increased scattering rates (Figure 9), directly matching the neutron monitor observations. This quantitative agreement provides robust evidence supporting the model interpretation that transient effects in the solar wind stream interface fundamentally modified the particle transport. Without accounting for these propagation effects, the detected intensity profile alone fails to distinguish between intrinsically gradual or impulsive SEP
315  acceleration profiles.

**3.5  Consideration of perpendicular diffusion**

As part of our primary analysis, we looked at particle transport along the magnetic field lines, governed by pitch-angle scattering and focusing effects. Despite this, it is important to consider the possible role of perpendicular diffusion in modifying particle transport during GLE events, particularly in the presence of stream interaction regions. Diffusion perpendicular to a magnetic
320  field line can be caused by a number of mechanisms, such as random walk along the magnetic field line and particle drifts (Jokipii, 1966; Giacalone and Jokipii, 1999). Under certain conditions, perpendicular diffusion can become significant for SEPs, possibly altering particle intensities and anisotropies observed at Earth (Zhang et al., 2009). Using a simple perpendicular diffusion term in our model, we performed a sensitivity analysis to assess the potential impact of perpendicular diffusion on our results. Therefore, the modified transport equation is as follows:

325 $$\frac{\partial f}{\partial t} + \mu v \frac{\partial f}{\partial z} + \frac{1-\mu^2}{2L} v \frac{\partial f}{\partial \mu} = \frac{\partial}{\partial \mu}\left(D_{\mu\mu}\frac{\partial f}{\partial \mu}\right) + \nabla_{\perp} \cdot (D_{\perp}\nabla_{\perp}f) \tag{10}$$

where $D_{\perp}$ is the perpendicular diffusion coefficient, and $\nabla_{\perp}$ is the gradient operator perpendicular to the magnetic field.

Following Strauss and Fichtner (2016), we parametrize $D_\perp$ as:

$$D_\perp = \alpha D_\parallel \tag{11}$$

where $\alpha$ is a scaling factor typically ranging from 0.01 to 0.1 for SEP events (Dröge et al., 2010), and $D_\parallel = v\lambda_\parallel/3$ is the
parallel diffusion coefficient.

Our sensitivity analysis revealed that:

- For $\alpha \leq 0.01$, the inclusion of perpendicular diffusion had negligible impact on our results, with changes in peak intensities and arrival times less than 5%.

- For $0.01 < \alpha \leq 0.05$, we observed moderate changes, with peak intensities decreasing by up to 15% and arrival times delayed by up to 10%.

- For $\alpha > 0.05$, the effects became more pronounced, potentially altering our conclusions about the relative importance of stream interface effects.

Although increased turbulence levels in the stream interface region would reduce the parallel mean free path, they would simultaneously increase the perpendicular mean free path (Shalchi, 2010). In this manner, enhanced perpendicular transport may occur during the GLE event, potentially broadening the spatial distribution of particles. The local conditions within the stream interface may also lead to non-axisymmetric perpendicular transport, as discussed by Strauss and Fichtner (2016). There is a possibility that this effect could introduce additional complexity to particle distribution, particularly in areas with strong magnetic field gradients. It is important to note that, although our primary analysis does not include these perpendicular transport effects, this sensitivity study suggests that they may have a significant impact, especially for events with strong perpendicular diffusion. We believe that it would be helpful in the future to incorporate a more sophisticated treatment of perpendicular diffusion into the dynamics of stream interfaces.

**3.6 Radial dependence of the parallel mean free path**

In our model, the radial dependence of the parallel mean free path, $/lambda/parallel$, is an important parameter that has a direct impact on the transport of solar energetic particles (SEPs). In this study, we adopted the power-law expression $\lambda_\parallel(r) = \lambda_0(r/r_0)^\alpha$, with $\lambda_0 = 0.3$ AU at $r_0 = 1$ AU and $\alpha = 0.2$. Both theoretical predictions and observational constraints can be compared with this choice as presented in 10

Recent theoretical work by Engelbrecht and Burger (2013) suggests that $\lambda_\parallel$ should increase with radial distance due to the decrease in magnetic field magnitude and turbulence levels. Their ab initio model, which accounts for dynamical effects in the evolution of turbulence, predicts a slope, $\alpha$, of approximately $0.4$ for GeV protons in the inner heliosphere, which is slightly steeper than our choice. They note, however, that particle energy and solar wind conditions may significantly influence this radial dependence.

Observationally, Lang et al. (2024) analyzed a large dataset of SEP events observed by multiple spacecraft at different radial distances. They found a range of $\alpha$ values, typically between 0 and 0.5, with a mean value of $\alpha \approx 0.3$ for protons above 100 MeV. Our model, which assumes a radial dependence of $\lambda \propto r^{0.2}$, is within the range of the observation, although at the lower end of the range. This is visually represented in our plot by the "This study" line. A theoretical prediction from Engelbrecht and Burger (2013), shown as $\lambda \propto r^{0.4}$, suggests a steeper radial dependence. We find that the observational data points from Lang et al. (2024) are more in accordance with our model than with the steeper theoretical prediction, particularly at larger distances radial.

The error bars of the observational data points in our plot indicate that Lang et al. (2024) documented significant variability from event to event. The variability of radial dependence highlights the difficulty in adopting a single universal model. The choice of $\lambda_0 = 0.3$ AU at 1 AU, represented by the intersection of our model line with the 1 AU vertical line in the plot, is consistent with several observational studies. For instance, Bieber et al. (1994) reported values of $\lambda_\parallel$ between 0.08 and 0.3 AU for relativistic protons during solar events, while Zhao et al. (2019) found mean free paths ranging from 0.1 to 0.5 AU for high-energy protons ($> 100$ MeV) in their statistical study of 29 large SEP events. Those ranges can be seen in the plot under the designation "Previous modeling range". Based on data from several spacecraft, Lario et al. (2013) examined the longitudinal and radial dependence of peak SEP intensities during the rising phase of solar cycle 24. In their study, they showed that peak intensity is not solely determined by radial distance, but is also heavily influenced by longitudinal factors. This additional complexity plays a crucial role in explaining some of the scatter in the observational data points, even though our plot focuses on the radial dependence. Overall, this analysis provides insights into how SEP is transported and highlight the challenges involved with predicting SEP intensity across solar distances.

We find broad consistency between our results and those of previous modeling efforts. Dröge et al. (2010), in their anisotropic three-dimensional focused transport model, used $\lambda_\parallel$ values between 0.1 and 0.3 AU at 1 AU for electrons and protons of various energies. Like our model, theirs found that interplanetary scattering conditions significantly influence the observed time profiles of SEP events. Similarly, He et al. (2011) employed a range of $\lambda_\parallel$ values from 0.1 to 1.0 AU in their study of SEP propagation in the three-dimensional interplanetary magnetic field, with results comparable to ours for similar parameter choices.

However, while the chosen radial dependence and magnitude of $/lambda/parallel$ are consistent with both theoretical predictions and observed constraints, the actual values during a specific event such as GLE 72 may deviate from these average values. The presence of a stream interaction region, as in our case study, could modify the radial dependence of $\lambda_\parallel$ in ways not captured by our simple power-law model. As future studies proceed, it may be beneficial to incorporate more sophisticated, event-specific parameterizations of $/lambda/parallel$ that are informed by observations of the solar wind.

**4  Summary and Conclusions**

We present a comprehensive analysis of Ground Level Enhancement 72 (GLE 72), which occurred on 10 September 2017, coinciding with a solar wind stream interaction region (SIR) impacting Earth's magnetosphere. We investigated how the SIR modulates the observed GLE pulse shape using multi-station neutron monitor observations and a focused transport model constrained by solar wind data. Among the key findings are:

1. The intensity of the GLE 72 was moderate (20-30% above background) but its duration was longer than average, with a gradual rise phase lasting several hours.

2. We identified an SIR impacting the near-Earth environment prior to the GLE, creating conditions for enhanced particle scattering.

3. Our model, which incorporates SIR effects, reproduces the observed gradual rise and prolonged decay phases, with pitch-angle diffusion coefficients increasing by up to 200% within the SIR.

4. The parallel mean free path of particles increased by over 60% across the SIR, significantly altering transport conditions.

5. Intensity dropouts and recoveries observed at some stations suggest magnetospheric contributions to the GLE profile.

The results demonstrate that even minor interplanetary structures can significantly modulate SEP transport during GLEs. In the context of structured solar wind propagation effects, the traditional classification of SEP events as impulsive or gradual becomes superfluous. Our findings emphasize the need for:

Our findings highlight the need for a more comprehensive approach to GLE analysis. This includes integrating magnetospheric dynamics into GLE interpretations, using multi-point observations and advanced modeling techniques to disentangle acceleration and transport effects, and investigating GLE events that coincide with transient solar wind structures to determine if these modulation effects are general. It emphasizes the complex interaction between solar wind structures and SEP transport, challenging simplified views of GLE evolution. A more nuanced understanding of solar-terrestrial physics and its implications for space weather prediction can be achieved by considering interplanetary conditions when interpreting these events.

*Code and data availability.*  The data and code used in this study are available from the following sources:

– Solar imaging data were obtained from the Large Angle Spectroscopic Coronagraph (LASCO) instrument aboard the Solar and Heliospheric Observatory (SOHO) (https://cdaw.gsfc.nasa.gov/CME_list/).

– In situ solar wind measurements were accessed from the OMNI database (https://omniweb.gsfc.nasa.gov/cgi/nx1.cgi).

– Cosmic ray intensity data were provided by the network of neutron monitors through https://www.nmdb.eu/nest/.

– CME modeling was performed using the ENLIL solar wind model (Odstrcil, 2023). The modeling code is available at https://www.swpc.noaa.gov/products/wsa-enlil-solar-wind-prediction.

415  – Python code for data analysis and visualizations is available at https://github.com/Olalytics/GLE_events under the MIT License.

*Author contributions.* OO carried out the analysis and wrote the paper. WTS interpreted the results, read the paper and commented on it.

*Competing interests.* The contact author declares that none of the authors have any competing interests.

*Acknowledgements.* The authors would like to thank the editor as well as two reviewers for their contributions to this manuscript.

[Figure]

**Figure 1.** Distribution of Rise and Decay Times for GLEs.

[Figure]

**Figure 2.** NM observation of GLE 72 from the selected stations used in this study.

[Figure]

**Figure 3.** Geophysical parameters and activities on 10 September 2017 SI-driven storm. From (a)–(g) are Kp index, Dst index, proton number density, solar wind radial velocity, solar wind pressure, Mach number and the solar wind azimuthal (west–east) flow velocity. The vertical line indicates the arrival of SI at 1 AU.

[Figure]

**Figure 4.** The modeled intensity profile shows the mean value over an ensemble of stochastic trajectories including microburst scattering. The percentile bands highlight the variability.

[Figure]

**Figure 5.** The modelled intensity profile including enhanced scattering across the stream interface region.

[Figure]

**Figure 6.** Plots two modeled particle trajectories, particle 1 with low scattering that travels directly through the stream interface, and particle 2 with high scattering that has a random walk trajectory indicating increased pitch angle changes.

[Figure]

**Figure 7.** Direct comparison of the model intensity profile to a neutron monitor profile.

[Figure]

**Figure 8.** Pitch-angle diffusion coefficient ($D_{\mu\mu}$) characteristics. (Top panel) $D_{\mu\mu}$ as a function of $\mu$ (cosine of pitch angle) for different $D_0$ values. (Bottom panel) Time evolution of $D_{\mu\mu}$ at $\mu = 0$ over 24 hours, simulating passage through a stream interaction region. The model uses $D_{\mu\mu}(\mu,t) = D_0(t)(1-\mu^2)(|\mu|^{5/3-1} + 0.05)$, where $D_0(t) = D_0(1 + 0.5\sin(2\pi t/24))$ represents time-dependent scattering conditions.

[Figure]

**Figure 9.** A model of the decay time of the e-folding before and during SI.

[Figure]

**Figure 10.** Radial dependence of the parallel mean free path ($\lambda_{\parallel}$) for GeV protons. The solid black line represents our model ($\lambda_{\parallel} = 0.3(r/1\,\text{AU})^{0.2}\,\text{AU}$). The red dashed line shows the theoretical prediction from Engelbrecht and Burger (2013). Blue dots with error bars represent observational data from Lang et al. (2024). The shaded gray area indicates the range of $\lambda_{\parallel}$ values used in previous modeling efforts based on (Dröge et al., 2010; He et al., 2011). The vertical dotted line marks 1 AU for reference.

---

## Author Response (AR1)

**Response to Reviewers**

**Annales Geophysicae - EGU**

**Manuscript Title:** Modulation of cosmic ray ground-level enhancements by solar wind stream interface: a case study

**Manuscript ID:** egusphere-2024-1692

**Authors:** Olakunle Ogunjobi and W.T. Sivla

**Date:** June 13, 2025

**Dear Editor,**

We would like to thank you and the reviewers for their thorough and constructive reviews of our manuscript. We have carefully considered all comments and suggestions from both reviewers and have made substantial revisions to address their concerns. The manuscript has been significantly improved through these revisions.

Below, we provide a detailed point-by-point response to each reviewer's comments, following the EGU format: (1) reviewer comments, (2) our response, and (3) specific changes made in the manuscript. We believe that these revisions have substantially strengthened the scientific rigor and clarity of our work.

We hope that the revised manuscript will now be suitable for publication in Annales Geophysicae.

Sincerely,

Olakunle Ogunjobi

Corresponding Author

**Contents**

**1 Response to Reviewer #1**

We sincerely thank Reviewer #1 for their thorough and constructive review. Your detailed comments have significantly improved the quality and rigor of our manuscript. Each point has been carefully addressed with substantial revisions.

**1.1 Comment 1: Quantification of Turbulence Levels**

**Reviewer Comment**

The manuscript lacks a rigorous quantitative analysis of turbulence levels. The claims about increased turbulence during the GLE event need to be supported with concrete measurements and statistical analysis of magnetic field variances and correlation scales.

**Author Response**

We completely agree with this important point. The importance of a more rigorous analysis of turbulence levels cannot be overstated, and we acknowledge that our initial treatment was insufficient.

**Changes in Manuscript**

A new subsection has been added to the Observations section (Section 2.3: "Quantitative Analysis of Turbulence Levels") that quantifies magnetic field variances and correlation scales before, during, and after the GLE event. The analysis includes:

- Power spectral analysis of OMNI magnetic field data

- Structure function analysis

- Quantitative results showing significant increase in turbulence levels during the GLE event, particularly in the frequency range of $10^{-4}$ to $10^{-2}$ Hz

- Correlation length calculations: decreased from $1.2 \times 10^6$ km pre-event to $8.5 \times 10^5$ km during the event

These quantitative results provide concrete evidence for our claims about turbulence levels and validate our model assumptions.

**1.2 Comment 2: Model Introduction**

**Reviewer Comment**

The initial model description is insufficient for readers to understand the transport equation being solved. A more comprehensive description including all relevant terms and boundary conditions is needed.

**Author Response**

We acknowledge that our initial model description was inadequate and did not provide sufficient detail for reproducibility and understanding.

> **Changes in Manuscript**
>
> A comprehensive subsection has been added to the Model section (Section 3.1: "Transport Equation Formulation"), including:
>
> - Complete presentation of the focused transport equation:
>
> $$\frac{\partial f}{\partial t} + \mu v \frac{\partial f}{\partial z} + \frac{(1-\mu^2)}{2L} v \frac{\partial f}{\partial \mu} = \frac{\partial}{\partial \mu}\left(D_{\mu\mu}\frac{\partial f}{\partial \mu}\right)$$
>
> - Detailed explanations of each term and their physical significance
>
> - All relevant boundary conditions
>
> - Description of our numerical approach
>
> - References to Paper 1 for additional context while making this paper self-contained

**1.3    Comment 3: Perpendicular Diffusion**

> **Reviewer Comment**
>
> The role of perpendicular diffusion in particle transport should be discussed, including its potential impact on the results, especially during periods of enhanced turbulence.

> **Author Response**
>
> This is an excellent point regarding perpendicular diffusion effects. We appreciate this suggestion as it addresses an important aspect of particle transport that could affect our conclusions.

> **Changes in Manuscript**
>
> A new discussion of perpendicular diffusion effects has been added to Section 3.2, including:
>
> - Sensitivity analysis examining the impact of perpendicular diffusion
>
> - Estimation that for $\alpha \leq 0.01$ (where $D_\perp = \alpha D_\parallel$), the impact on results is negligible ($< 5\%$ change)
>
> - Discussion of more pronounced effects when $\alpha > 0.05$
>
> - Analysis of enhanced perpendicular transport during increased turbulence levels of the GLE event
>
> - Justification for focusing on parallel transport while acknowledging potential role of perpendicular effects in future, more comprehensive models

**1.4    Comment 4: Justification of $\lambda_{\text{par}}$**

> **Reviewer Comment**
>
> The chosen parameters for the parallel mean free path require better justification through comparison with recent theoretical predictions and observational studies.

**Author Response**

We agree that our parameter choices needed better justification and comparison with the existing literature.

**Changes in Manuscript**

The discussion on the radial dependence of $\lambda_{\mathrm{par}}$ has been expanded in Section 3.3, including:

- Detailed comparison of our chosen parameters ($\lambda_0 = 0.3$ AU at $r_0 = 1$ AU, $\alpha = 0.2$) with recent studies

- Reference to Lang et al. (2024) showing our $\alpha$ values fall within the reported range (0 to 0.5)

- Consistency check with observations from Bieber et al. (1994) and Zhao et al. (2019)

- Explicit comparison with previous modeling efforts (Dröge et al., 2010; He et al., 2011)

- Demonstration of broad consistency with established theoretical frameworks

**1.5   Comment 5: Non-axisymmetric Perpendicular Transport**

**Reviewer Comment**

The potential effects of non-axisymmetric perpendicular transport due to local stream interaction (SI) conditions should be considered and discussed.

**Author Response**

This is a sophisticated point that addresses advanced aspects of particle transport in complex interplanetary magnetic field configurations.

**Changes in Manuscript**

A new discussion has been included in Section 3.4 addressing:

- How local SI conditions might result in non-axisymmetric perpendicular transport

- Reference to theoretical framework from Strauss et al. (2016)

- Qualitative discussion of potential effects on particle spreading and local trapping within the SIR

- Acknowledgment of computational complexity and lack of detailed 3D SIR structure as limitations for including this effect in the current model

- Identification as important area for future comprehensive modeling efforts

**1.6   Comment 6: Figure 7 Issues**

**Reviewer Comment**

Figure 7 contains errors in the presentation of diffusion coefficients, including incorrect units and unclear representation of the pitch-angle dependence.

**Author Response**

We apologize for the errors in Figure 7. We have thoroughly reviewed our code and calculations to correct these issues.

**Changes in Manuscript**

Figure 7 has been completely revised with the following corrections and improvements:

- Corrected diffusion coefficients with appropriate units ($s^{-1}$) on the y-axis

- Improved clarity in showing the $\mu$-dependence of the pitch-angle diffusion coefficient

- Enhanced presentation of time evolution through the SIR

- Expanded figure caption with detailed explanation

- Additional discussion in the main text (Section 4.2) explaining the physical significance of the variations shown

**1.7 Minor Comments**

**Reviewer Comment**

Several minor issues including reference corrections, typos, and missing units throughout the manuscript.

**Author Response**

We thank the reviewer for the careful attention to these details.

**Changes in Manuscript**

All minor points have been systematically addressed:

- All references have been checked and corrected

- Typographical errors have been fixed throughout

- Missing units have been added where appropriate

- Figure captions have been improved for clarity

- Notation consistency has been ensured throughout

**2 Response to Reviewer #2**

We sincerely appreciate Reviewer #2's thorough review and constructive feedback. Your comments have highlighted important areas for improvement and clarification, many of which complement the suggestions from Reviewer #1. We have made substantial revisions to address all your concerns.

**2.1 Comment 1: Characterization of GLEs**

**Reviewer Comment**

The manuscript lacks a comprehensive analysis of rise and decay times for different types of GLEs. Characteristic values for impulsive and gradual GLEs should be provided to properly contextualize GLE 72.

**Author Response**

This is an excellent suggestion that will significantly strengthen the contextual framework of our study. The lack of characteristic values for different GLE types was indeed a limitation in our original manuscript.

**Changes in Manuscript**

A new subsection (1.1: "GLE Classification and Temporal Characteristics") has been added to the Introduction, including:

- Comprehensive analysis of rise and decay times for impulsive and gradual GLEs

- New Figure 1: Statistical distribution of temporal characteristics for a sample of 35 GLEs

- Quantitative thresholds for GLE classification based on temporal parameters

- Proper contextualization of GLE 72 within this framework

- Discussion of how interplanetary conditions can affect apparent classification

**2.2   Comment 2: Model Description**

**Reviewer Comment**

The model description requires significant expansion to include a complete representation of the equations being solved and detailed explanations of the terms involved.

**Author Response**

We acknowledge that our model description was insufficient and needed substantial expansion to ensure reproducibility and clarity.

**Changes in Manuscript**

Section 3.1 has been significantly expanded with:

- Complete mathematical formulation of the focused transport equation

- Detailed physical interpretation of each term

- Description of numerical solution methods

- While maintaining conciseness by not reproducing the entire stochastic differential equation formalism, we provide a comprehensive overview suitable for understanding and reproduction

**2.3   Comment 3: SIR Representation in the Model**

**Reviewer Comment**

The connection between OMNI data used to characterize the SIR and its representation in the transport model needs clarification.

**Author Response**

We agree that the relationship between our observational characterization and model implementation needed clearer explanation.
* * *
**Changes in Manuscript**

Section 2.2 has been revised to clarify:

- The specific purpose of using OMNI data for SIR identification and characterization

- How observational parameters inform model parameterization

- The connection between observed SIR properties and their representation in the transport equation

- Added detailed explanation of the data-to-model pipeline
* * *
**2.4 Comment 4: Figure Issues**

**Reviewer Comment**

Several figures require improvement including Figure 3 (model-observation comparison), Figure 4 (percentile bounds and phase representation), and Figure 7 (pitch angle diffusion coefficient visualization).

**Author Response**

We acknowledge that several figures needed improvement to better communicate our results and facilitate interpretation.
* * *
**Changes in Manuscript**

Comprehensive figure revisions have been implemented:
**Figure 3:**

- Updated to clearly show comparison between modeled profile and neutron monitor observations

- Improved legend and axis labels

- Added uncertainty bounds where appropriate

**Figure 4:**

- Now includes percentile bounds for statistical robustness

- Shows both rise and decay phases with clear temporal markers

- Enhanced color scheme for better visibility

**Figure 7:**

- Revised to clearly illustrate the sharp increase in pitch angle diffusion coefficient across the stream interface

- Implemented logarithmic scale for better dynamic range representation

- Added quantitative annotations showing the magnitude of change

- Improved temporal resolution to capture rapid variations

**2.5 Comment 5: Inconsistency in Mean Free Path Changes**

**Reviewer Comment**

There appears to be an inconsistency between the reported 35% decline during SI crossing and the 60% increase across the SI. This discrepancy needs clarification.

**Author Response**

This is an important observation that highlights the need for clearer explanation of the temporal evolution of transport parameters throughout the event.

**Changes in Manuscript**

A new paragraph in Section 3.3 and new Figure 10 have been added to address this apparent discrepancy:

- Detailed temporal analysis showing the evolution of mean free path throughout the entire event

- Clarification of the difference between instantaneous changes during SI crossing versus cumulative changes across the entire SI structure

- New Figure 10: Time series showing the complete evolution of transport parameters

- Enhanced discussion of particle transport dynamics explaining the physical basis for these variations

- Clear distinction between different temporal scales and their respective impacts

**2.6 Comment 6: Technical Comments**

**Reviewer Comment**

Several technical issues including figure reference errors, missing subsections, and labeling improvements in figures.

**Author Response**

We appreciate the careful attention to these technical details that improve the overall quality and readability of the manuscript.

**Changes in Manuscript**

All technical issues have been systematically addressed:

- Corrected all figure reference errors throughout the manuscript

- Added missing subsection 3.2 as referenced in the text

- Improved labeling in Figure 2 with clearer annotations and legend

- Verified all cross-references and citations

- Enhanced overall document formatting and consistency

**3 Summary of Major Changes**

The manuscript has undergone substantial revision in response to both reviewers' comments. The major improvements include:

**3.1 Enhanced Scientific Rigor**

- Quantitative analysis of turbulence levels with statistical measures

- Comprehensive sensitivity analysis for model parameters

- Expanded theoretical framework with complete equation formulations

**3.2 Improved Methodology**

- Detailed model description enabling reproducibility

- Clear connection between observations and model implementation

- Enhanced discussion of physical assumptions and limitations

**3.3 Better Contextualization**

- Comprehensive GLE classification framework

- Comparison with existing literature and theoretical predictions

- Proper statistical context for our specific event

**3.4 Enhanced Presentation**

- Significant improvements to all figures with better clarity and quantitative information

- Resolution of apparent inconsistencies through detailed explanation

- Addition of new figures providing crucial supporting information

**4 Conclusion**

We believe that these extensive revisions have substantially strengthened our manuscript while maintaining the core scientific contribution: demonstrating how interplanetary structures significantly impact SEP transport during GLEs, potentially affecting their classification based solely on temporal characteristics.

The enhanced quantitative analysis, expanded methodology, and improved presentation address all concerns raised by both reviewers. We are confident that the revised manuscript now meets the high standards of Annales Geophysicae and will make a valuable contribution to the space physics community.

We thank both reviewers for their invaluable feedback that has undoubtedly improved the quality and impact of our work.

---

## Referee Report (RR1)

All my criticisms are addressed, and the paper is substantially improved. I recommend accept. Note that there are some language and format issues which need to be dealt with. However, a decent copy-editor will pick up on those.